

# Traffic flow simulation of modified cellular automata model based on producer-consumer algorithm

Xuefeng Deng,  Yi Shao,  Jiaxin Song and  Hui Wu

College of Information Science and Engineering, Shanxi Agricultural University, Taigu, Shanxi, China

## ABSTRACT

With the rise of new technologies such as the Internet of Vehicles and the Internet of Things, research on the intelligent connected vehicle has become a hot topic in contemporary times. The modeling and simulation of traffic flow are mainly used to analyze the characteristics of traffic flow and study the formation and dissipation mechanism of traffic congestion to better guide the real traffic. Cellular automata are suitable for the simulation of complex giant systems. Because of the randomness and discreteness of vehicle driving, cellular automata are often used to model and analyze traffic flow. This article mainly studies the traffic flow formed by intelligent connected vehicles. Based on the traditional NaSch model, the producer-consumer algorithm is introduced to form a multi-buffer vehicle information access mode, and an improved cellular automata model with random updates is constructed. The simulation results show that the improved cellular automata model improves the traffic congestion significantly compared with the original NaSch model in the intelligent network environment, which is consistent with the actual traffic situation. Therefore, the algorithm proposed in this article can effectively simulate the traffic flow characteristics of intelligent connected vehicles, and provide a theoretical basis for solving traffic problems.

## INTRODUCTION

In recent years, with the rapid development of social economy, the number of urban car ownership is on the rise. However, the expansion of urban roads and the construction of transportation infrastructure are far from being able to meet the substantial increase in the number of cars, thus causing traffic problems such as traffic congestion (*Yao et al., 2021*). In order to alleviate urban traffic congestion and improve road traffic efficiency, intelligent networked vehicles emerge as the times require (*Chu et al., 2021*).

At present, the proportion of intelligent connected vehicles in urban traffic is increasing day by day (*Wei et al., 2020*). Scholars at home and abroad have researched the traffic flow simulation of new intelligent connected vehicles. *Zhang et al. (2021)* constructed a mixed traffic flow model with intelligent connected vehicles and analyzed the factors influencing the stability of traffic flow. *Liu (2021)* studied the traffic flow characteristics of intelligent connected vehicles under different proportions in mixed traffic flow and showed that

Corresponding author
Xuefeng Deng, dxf@sxau.edu.cn

with the increase of intelligent connected vehicles, the speed and density of traffic flow increased to a certain extent. *Pan & Lin (2021)* constructed a microscopic traffic flow model to analyze the behavioral decisions of intelligent connected vehicles at no-signal intersections. There are still many deficiencies in the research of urban road traffic flow model under an intelligent network environment, which can not fully show the traffic flow characteristics of new intelligent connected vehicles. Therefore, the traffic flow modeling analysis of intelligent connected vehicles still needs further research.

The discreteness of cellular automata model in time, space and state can simplify complex traffic flow problems (*You & Zhou, 2021*), so it is widely used in the study of traffic flow and becomes a powerful tool for studying road traffic flow problems. *Matcha et al. (2020)* summarized the car following model and simulation framework under mixed traffic flow, constructed 2D driver behavioral models, discussed the interaction between vehicles under mixed traffic flow, and provided a scheme to alleviate traffic congestion. *Jiang & Wu (2002)* used cellular automata to simulate synchronous traffic flow and analyze the congestion, dissipation, and other characteristics of traffic flow. Through the difference in acceleration and deceleration behavior under different driving conditions, the Jiang and Wu model was improved by *Tian et al. (2009)*, and the deceleration difference was used to reproduce the synchronous flow. *Tian et al. (2021)* discussed the cellular automata model for studying synchronous flow in traffic flow. And the importance of randomness in cellular automata model is proposed. *Hua et al. (2020)* studied driving behaviors on roads with three-dimensional facilities such as toll stations and construction areas and proposed a traffic flow model with changes in cellular space width. *Wahle et al. (2001)* proposed a theoretical model of online simulated road traffic flow that can reproduce traffic quality and carried out dynamic management of the road traffic network. *Tong & Feng (2020)* evaluated the cellular automata models in the past 20 years, summarized the specific evaluation indicators, and analyzed the simulation quality and reproduction reliability of the models.

For intelligent networked vehicles that need to share information in a large range, the traditional traffic flow research based on cellular automata cannot show the randomness of information transmission in information sharing due to its parallelism. This article proposes an improved cellular automata model based on the producer–consumer algorithm to study the traffic flow characteristics of intelligent connected vehicles.

The main contributions of this article are as follows: (1) The idea of the producer–consumer algorithm is introduced into the cellular automata model, which makes the simulation of road traffic flow more in line with the information sharing mechanism of vehicles under the condition of intelligent network connection. (2) Effectively analyze the application of intelligent network to traffic flow problems, and verify the analysis results that intelligent networked vehicles can improve road traffic congestion. (3) Through the analysis of the simulation results, theoretical guidance is provided for increasing the free flow area in road traffic, increasing the speed of vehicles, and accelerating the dissipation phenomenon of congestion.

# RELATED WORK

Traffic flow theory appeared in the 1930s and was mainly studied using the methods of probability theory. *Kinzer (1933)* introduced probability theory to highways in 1933, and discussed the application of Poisson distribution in traffic problems. In 1935, *Greenshields (1935)* conducted research on traffic capacity, which constructed a mathematical model through the methods of probability theory and mathematical statistics, and the relationship between flow and speed was proposed. In 1936, *Adams (1936)* treated road traffic as a random sequence and published numerical examples. In 1947, *Greenshields (1947)* used Poisson's law to analyze the traffic performance of urban intersections. In the 1950s, as the industry developed, probabilistic methods could not represent the interactions between vehicles. Car following theory, traffic wave theory and vehicle queuing theory appeared one after another. In December 1959, the convening of the International Symposium on Traffic Flow Theory marked that the traffic flow theory entered a period of rapid development. Scholars mainly conduct research on traffic flow characteristics, traffic flow models, and traffic flow simulation. Due to the complexity and dynamics of the traffic system (*Esser & Schreckenberg, 1997*), the use of simulation technology to study traffic behavior can more intuitively reveal the basic laws and formation mechanisms of traffic flow phenomena (*Monteil et al., 2014*). According to the different simulation objects, traffic simulation can be divided into microscopic, mesoscopic and macroscopic traffic flow simulation (*Burghout, Koutsopoulos & Andreasson, 2005*). The microscopic traffic flow simulation studies the individual behavior of vehicles, mainly represented by the car following model and the cellular automaton model (*Ming et al., 2012*). The real traffic system has a large scale and a wide variety. The cellular automata model has developed rapidly in the field of traffic flow simulation with the progress of computer application technology due to its simple structure and fast calculation (*Brilon & Wu, 1999*). Therefore, cellular automata has a wide application prospect in the field of transportation.

In the early 1950s, Von Neumann proposed the theory of cellular automata to solve the problem of self-replication of machines (*Neumann, 1951*). In the 1970s, John Horton Conway created a "game of life" that simulates the characteristics of natural life by formulating three simple rules of survival (*Martin, 1970*). In 1986, Cremer and Ludwig first applied cellular automata theory to the simulation modeling of traffic flow, and achieved rapid development in the field of traffic (*Cremer & Ludwig, 1986*). In the 1980s, Wolfram conducted a comprehensive study of the models produced by the 256 rules of one-dimensional cellular automata. Among them, the cellular automata model of Rule 184 was applied in the field of traffic flow and became the basic model to simulate complex traffic flow phenomenon (*Wolfram, 1984*). In 1992, Nagel and Schreckenberg introduced cellular automata into highway modeling and proposed the classic NaSch model (*Nagel & Schreckenberg, 1992*). NaSch model formulates four evolution rules of acceleration, deceleration, random slowing down and position update to realize the simple simulation of single-lane traffic flow. As a classic case of cellular automata applied to two-lane traffic flow, STNS(symmetric two-lane Nagel–Schreckenberg) model was introduced by *Chowdhury, Wolf & Schreckenberg (1997)*. The STNS model adds two lane-changing rules based on the

**Table 1   Summary of cellular automata models.**

| Application | Model | Methods for improvement | Advantage | Disadvantage |
|---|---|---|---|---|
| Study on the model of expressway traffic | NaSch model | Random slowing down probability is introduced and maximum speed is no longer 1 | The rules are simple and can describe the actual traffic phenomenon | The simulation of complex traffic flow phenomenon cannot be completely realized |
| | TT model | Add slow start rule | Metastable and hysteresis phenomena can be obtained | Traffic flow is lower than NaSch model |
| | FI model | Improve acceleration rule | Easy to analyze and study | There is a big gap between rules and reality |
| | KKW model | Consider the speed effect of the front vehicle | There is a new phase called synchronous flow | The influence of the car behind on the car in front is not considered |
| Study on the model of urban road network traffic | BML model | The first two-dimensional traffic flow cellular automata model | The rules are simple and reveal some basic characteristics of urban traffic flow | Unable to accurately describe some problems in real urban road network traffic |
| | Gu proposed extended BML model | Change 2D uniform mesh to non-uniform mesh | The traffic lights at each intersection can freely choose the cycle and phase | The model needs to be further refined |
| | ChSch model | Applying NaSch model rules to the vehicle update process in BML model | Describes the movement of traffic flow on a road between adjacent intersections | A cell represents an intersection |
| | Freund proposed extended BML model | Change one-way traffic to two-way traffic | Extend the vehicle movement direction to four | The model can be further expanded |

NaSch model, which are the driver's lane-changing motivation and the safety distance between vehicles, which can be used to simulate lane-changing behavior in multi-lane traffic flow. After that, *Knospe et al. (2002)* proposed an asymmetric lane-changing rule by extending the expected driving single-lane model, and an asymmetric two-lane cellular automaton model was constructed. These are all studies of discrete cellular automata. In 2012, a continuous cellular automaton based on fuzzy reasoning was proposed by *Yeldan et al. (2012)*. The cellular automata theory has been keeping up with the changes of the times to improve the traffic flow model.

Based on the 184 model, the applications of cellular automata models in traffic flow can be divided into two categories. The research on urban road traffic flow represented by NaSch model and the research on urban traffic network represented by BML model are respectively. Their extended models are shown in Table 1.

# THEORETICAL BASIS

## Cellular automata theory

A cellular automaton is a dynamic system that evolves in the discrete-time dimension in the cellular space composed of discrete and finite-state cells according to certain local rules (*Chai et al., 2015*). Cellular automata are mainly composed of cellular, cellular space, neighbor, and cellular state evolution rules, which can simulate many complex giant

systems in real life according to simple evolution rules (*Valente et al., 2018*). Cellular can also be called cell, unit, which is the most basic component of cellular automata. Cell space is the collection of all grid points in the cell space. Among them, the cellular space of two-dimensional cellular automata can be divided into a triangular grid, quadrilateral grid, and hexagonal grid according to the geometric space division. The local cell space searched by the central cell to update its status is called the neighbor cell of that cell. One-dimensional cellular automata can determine neighbor cells according to a radius. The common neighbor types of two-dimensional cellular automata include the Von Neumann type, Moore type, and extended Moore type. The evolution rule of cellular state is the dynamic function that the central cellular determines the state at the next moment according to its neighbor cellular and its state. The rationality of the setting of evolution rules determines whether the cellular automata model can objectively show the essential characteristics of the system, which is the core of the whole cellular automata model construction.

## Basic cellular automata model—NaSch model

In NaSch model, $1 * n$ squares are used to represent driving roads and form cellular space. The current cell takes the left and right cells as the neighbor cell. The current cell can be empty, or it can be occupied by a vehicle. When occupied by a vehicle, the vehicle speed is between 0 and $V_{max}$. NaSch model added stochastic factors based on model 184, which is used to simulate various influencing factors in the driving process, and random slowing down probability $P$ is introduced. There are four evolution rules at the moment of cellular $t \sim t+1$, which are:

**Accelerate:**

$V_n(t+1) \rightarrow \min\{V_n(t)+1, V_{max}\}.$

Indicates that the driver always wishes to travel at maximum speed.

**Decelerate:**

$V_n(t+1) \rightarrow \min\{V_n(t), D_n\}.$

The driver slowed down to avoid a rear-end collision.

**Random slowing down (With probability $P$):**

$V_n(t+1) \rightarrow \max\{V_n(t)-1, 0\}.$

Driver deceleration measures caused by various uncertainties.

**Location Update:**

$X_n(t+1) \rightarrow X_n(t) + V_n(t+1).$

The car continued to drive at the updated speed.

Where $X_n$ represents the position of vehicle $n$, $V_n$ represents the speed of vehicle $n$. $D_n$ represents the distance between vehicle $n$ and vehicle $n+1$, $Dn = X_{n+1} - X_n - L$, where $L$ represents the length of the vehicle.

## Producer–consumer model

The producer–consumer problem (*Zhang et al., 2017*) was first proposed by *Dijkstra (1965)*. Producers and consumers share a buffer concurrently for the same period, and producers send the data they produce into the buffer for consumption by consumers. The consumer fetches data from the buffer and frees the buffer. When a producer produces data, if the buffer is full, it needs to wait for the consumer to release an empty buffer before it can continue to put data into the buffer. If the buffer is empty, the consumer must wait for the producer to put data into the buffer before continuing to consume.

The traditional producer–consumer model is mainly used to solve the problem of multi-thread concurrent execution in the operating system. In recent years, the technology of the Internet of Vehicles and the Internet of Things have developed rapidly. The producer consumer model is introduced into the modeling of multi-vehicle intelligent networking model optimization (*Vazquez-Lopez et al., 2021*) to solve the problems of information sharing and vehicle positioning between intelligent connected vehicles, which is of great significance to the research of intelligent connected vehicle traffic flow in real life.

## Improved cellular automata model

Intelligent connected vehicles are equipped with multiple types of sensors, which are used for large-scale information communication between vehicles (*Limbasiya & Das, 2019*). The driving information update of intelligent connected vehicles on the road is not carried out at the same time due to the influence of the distance difference between the vehicle in front, network quality, the driver's psychological state, driving habits and other factors. The NaSch model can simulate the basic acceleration and deceleration operations of vehicles, simulate the phenomenon of random slowing down caused by external factors, and study the formation and dissipation mechanism of traffic flow congestion. However, due to the concurrent execution mechanism of the cellular automata model, the random sharing of information between intelligent connected vehicles cannot be simulated.

The idea of sharing information with multiple buffers in the producer–consumer model can extend the single buffer mode used for information processing in traditional cellular automata models (such as NaSch model), and provide a solution to the problem of random information sharing among intelligent connected vehicles. Based on NaSch model, the location information and speed information of intelligent connected vehicles are stored in a multi-buffer group. Information sharing between vehicle sensors in the same period is stored randomly according to the sequence of data generation. After the data is passed into the buffer space, the following vehicle accelerates, decelerates, randomly slows down, and updates its position according to the shared information of the preceding vehicle. After data update, information is passed into the buffer space, forming a closed-loop of information sharing. The order of information access is related to the information processing of intelligent connected vehicles and has randomness. The sequential access mode of the previous cellular automata model is changed, and an improved cellular automata model with multi-buffer random access without putting back based on the producer–consumer algorithm is constructed. The model is used to simulate and analyze the traffic flow of intelligent connected vehicles with sensors, which can effectively show the situation of

**Table 2  Improved cellular automata algorithm.**

| Improved cellular automata algorithm |
| --- |
| 1: Initialize road parameters |
| 2: index=randperm(i);        %i represents the position of the first car. |
| %The randperm function is used to randomly extract integers between 1 and i without putting back. |
| 3: for j=1:index        %Loop iteration |
| 4:      v(i-j+1)=min(v(i-j+1)+1,vmax);        %Acceleration process |
| 5:      v(i-j+1)=min(v(i-j+1),d);        %Deceleration process |
| 6:      v(i-j+1)=randslow(v(i-j+1));        %Random slowing of vehicles |
| 7:      z(i-j+1+new_v)=1;        %Location update |
| 8:      v(i-j+1+new_v)=new_v;        %Velocity update |
| 9: end |

**Table 3  Simulation parameters of experimental model.**

| Parameters | NaSch model | Improved model | Model considering driver factors | Improved model considering driver factors |
| --- | --- | --- | --- | --- |
| Cell space (cells) | 1*1000 | 1*1000 | 1*1000 | 1*1000 |
| Initial number of vehicles (cells) | 200 | 200 | 200 | 200 |
| Initial vehicle speed (cells/s) | 1 | 1 | 1 | 1 |
| Maximum speed (cells/s) | 5 | 5 | 5 | 5 |
| Acceleration (cells/s$^2$) | 1 | 1 | 1/2 | 1/2 |
| Iterations (time) | 1000 | 1000 | 1000 | 1000 |
| Random moderation probability (%) | 0.3 | 0.3 | 0.3 | 0.3 |
| Whether the vehicle update is random | No | Yes | No | Yes |

information processing error updates of intelligent connected vehicles in the same period in real life. The problem of updating traditional cellular automata data at the same time is solved, which is consistent with the randomness of actual road traffic flow.

The algorithm in the case of the improved cellular automata model can be summarized as shown in Table 2.

## RESULTS AND DISCUSSIONS

The initialization of simulation parameters is shown in Table 3. The experimental road is a single lane with a total length of 1,000 cells, and 200 vehicles are randomly distributed on the road in the initial state. Assuming the length of the vehicle is 1 cell, the maximum speed is 5 cells/s. At the start of the simulation, the vehicle is traveling at a speed of 1 cell/s. In the traffic flow graph, the white square represents a vehicle at the position, and the black square represents an empty cell, that is, there is no vehicle at the position. In this experiment, the simulation step size is set to 1000 steps, and the opening boundary condition is used.

Figures 1A–1D represent the space–time diagrams of the NaSch model, the improved cellular automata model, the model considering the driver factor, and the improved model considering the driver factor, respectively. The abscissa represents the spatial position of the cell, and the ordinate represents the simulation time. Select the most representative

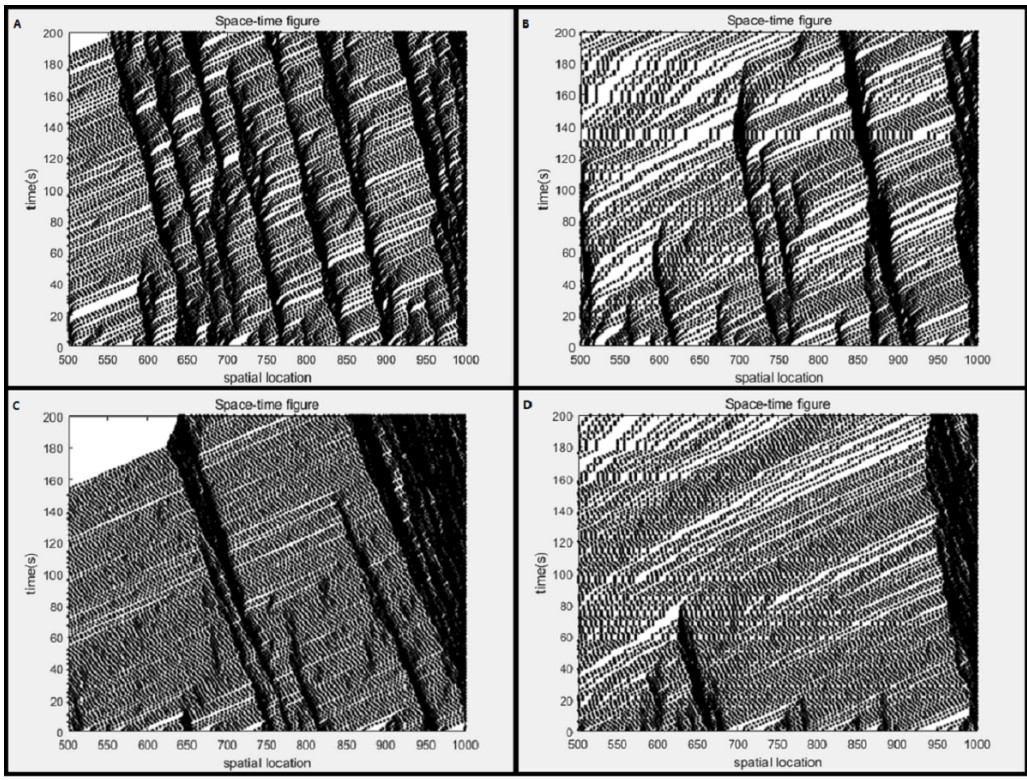

**Figure 1** **Space-time diagram.** (A) NaSch model; (B) improved cellular automata model; (C) model considering driver factors; (D) improved model considering driver factors.

500~1,000 spatial positions and 0~200s time cells to form the space–time map of the vehicle for discussion. The white square in the figure indicates that there is no car at the location, and the black square indicates that there is a car at the location. The driving direction of the vehicle is from left to right, and the time goes from bottom to top. As shown in Fig. 1A, the NaSch model shows that the traffic congestion gradually propagates upstream of the road over time, and there is a large number of wide congestion bands. The improved cellular automata model after adding the producer–consumer concept changes the rules of sequential update of vehicles on the road. And the position and speed of the vehicle are updated in a random form, which is more in line with the reality that the vehicle moves forward randomly. As shown in Fig. 1B, as time goes by, the number of blocking strips decreases and the width narrows, the traffic congestion on the upstream side of the road gradually dissipates, and the free flow area in the lane becomes larger, and there is no large-scale congestion area. Compared with the NaSch model, the improved cellular automata model significantly reduces the number of congestion zones, and the traffic congestion situation is greatly improved. It can be seen that the improved cellular automata model can effectively alleviate the traffic congestion on the road.

When the current vehicle speed is low, the driver tends to increase the acceleration with a high probability, thereby rapidly increasing the driving speed of the vehicle. At the

same time, some drivers tend to maintain the existing acceleration and gradually increase the speed. Based on this phenomenon, a new model is constructed by adding conditions to the classical NaSch model, which is called the model considering the driver factor, which considers the acceleration changes of the vehicle and integrates human behavior with a fuzzy variable. As shown in Fig. 1C, the congestion spreads upstream of the road over time. A wide blocking strip is gradually formed in front of the road and tends to expand. The space–time diagram of the improved model considering the driver factor after adding the producer–consumer concept is shown in Fig. 1D. Compared with Fig. 1C, the traffic congestion dissipates significantly over time, the number of blocking strips on the road decreases, and the width of the blocking strips narrows. The comparison shows that the improved model considering the driver factor can accelerate the dissipation of the congested area and expand the free flow area on the road.

Figures 2A–2D represent the flow-density diagrams of the NaSch model, the improved cellular automata model, the model considering the driver factor, and the improved model considering the driver factor, respectively. The NaSch model is shown in Fig. 2A, when the vehicle density increases, the traffic flow also increases. When the density increases to about 0.165, the flow reaches the maximum value of 0.49, that is, the optimal density is 0.165. Before reaching the optimal density, the traffic flow on the road is generally in a free-flow state. After reaching the optimal density, the traffic flow changes from a free-flow state to a blocked state, and the traffic is congested and the traffic flow decreases. The flow-density diagram of the improved cellular automata model is shown in Fig. 2B. Compared with Fig. 2A, the optimal density value of the improved cellular automata model is the same as that of the NaSch model. The improved cellular automata model is stronger in overall compactness of the image, and the traffic flow value before reaching the optimal density value is higher than that in the NaSch model. Therefore, compared with the NaSch model, the improved cellular automata model has higher vehicle velocity, less traffic congestion, less metastable traffic flow, and better road traffic conditions.

The flow-density diagram of the model considering the driver factor is shown in Fig. 2C. Before reaching the maximum traffic flow, with the increase of vehicle density, the traffic flow showed an upward trend. After that, with the increase of vehicles, the traffic congestion situation intensified, and the traffic flow showed a downward trend. When the vehicle density is about 0.17, the traffic flow reaches a maximum value of about 0.58, that is, the optimal density is 0.17. The flow-density diagram of the improved model considering the driver factor is shown in Fig. 2D. When the vehicle density is about 0.155, the traffic flow reaches a maximum value of about 0.57, that is, the optimal density is 0.155. The improved model after the introduction of the producer–consumer concept reduces the optimal density value and increases the overall traffic flow compared to Fig. 2C. The improved model accelerates the dissipation of traffic congestion and the expansion of free-flow areas on the road.

Build a thermodynamic diagram based on the speed of vehicles on the road. In this experiment, the vehicle speed is divided into six levels from 0 to 5, in which the yellow is the maximum speed of 5cells/s, and the blue is the stationary state, that is, 0cells/s. The abscissa of the thermodynamic diagram represents the spatial position of the cell, and the ordinate

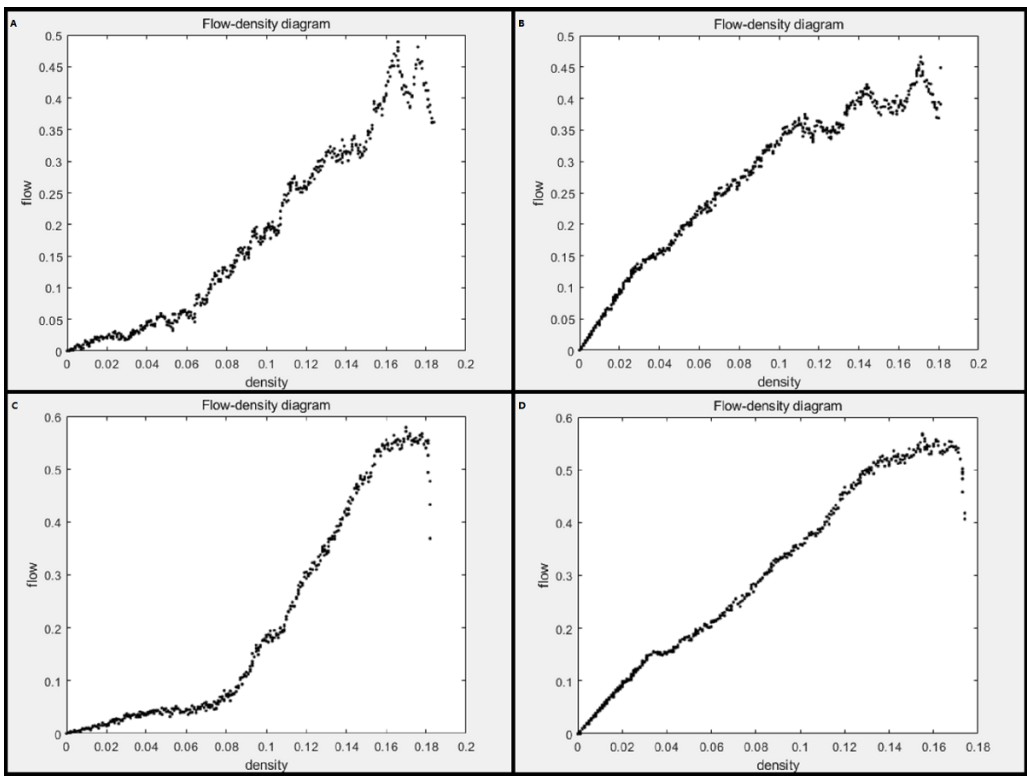

**Figure 2** **Flow-density diagram.** (A) NaSch model; (B) improved cellular automata model; (C) model considering driver factors; (D) improved model considering driver factors.

represents the simulation time. The driving direction of the vehicle is from left to right, and time moves in a top-to-bottom direction. Select the space–time region composed of 700-1000 spatial positions and 0-300s time cells for thermodynamic diagram analysis.

Figures 3A–3D represent the thermodynamic diagrams of the NaSch model, the improved cellular automata model, the model considering the driver factor, and the improved model considering the driver factor, respectively. The NaSch model is shown in Fig. 3A, with the passage of time, the vehicles on the road show a state of low speed, the vehicles stop and go, and the congestion increases. The thermodynamic diagram of the improved cellular automaton model is shown in Fig. 3B, with the passage of time, the blocking bars are reduced, the number of vehicles stop and go is reduced, and the number of vehicles traveling at the maximum desired speed of 5cells/s is increased. Compared with Fig. 3A, the improved cellular automata model improves road congestion, the area of the vehicle free flow state is increased, and most vehicles are in a high-speed driving state. Therefore, the improved cellular automata model can effectively dissipate traffic jams, improve the overall flow speed of the road, and meet the needs of drivers who always expect to travel at the maximum speed.

The thermodynamic diagram of the model considering the driver factor is shown in Fig. 3C. With the passage of time, the free flow area on the road decreases, traffic congestion increases, and low-speed vehicles gradually increase. The thermodynamic diagram of the

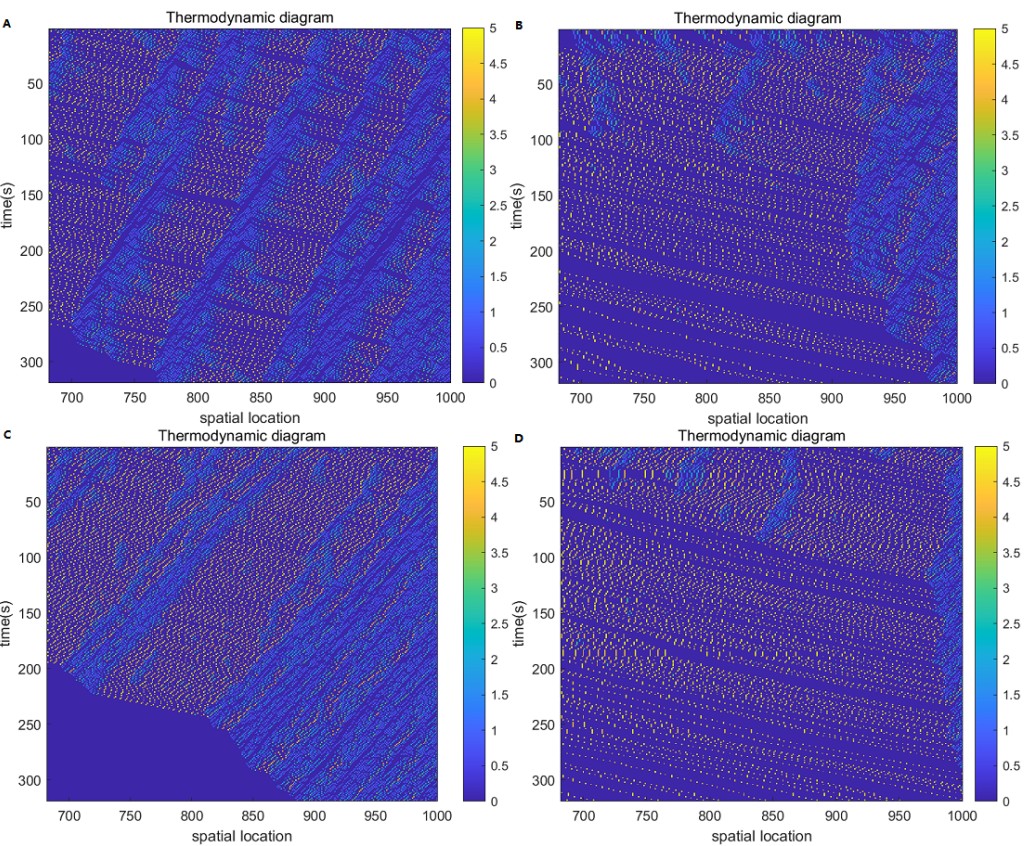

**Figure 3   Thermodynamic diagram.** (A) NaSch model; (B) improved cellular automata model; (C) model considering driver factors; (D) improved model considering driver factors.

improved model considering the driver factor is shown in Fig. 3D. With the passage of time, the free flow area of the improved model increases, traffic congestion is greatly improved, and most vehicles on the road travel at high speeds. Compared with Fig. 3C, the improved model has fewer stationary vehicles, more high-speed vehicles, and the overall flow velocity of the road is improved. This satisfies the driver's need to always expect to travel at the maximum speed, and improves the driver's driving experience on the road.

## CONCLUSIONS

Based on the NaSch model, this article introduces the producer–consumer model to improve it. According to the experimental results, the improved cellular automata model is more consistent with the driving conditions of vehicles on the road in real life. According to the analysis of experimental data, it shows that the improved cellular automata model can effectively ease road congestion and accelerate the dissipation of traffic jam. Compared with NaSch model, traffic flow and driving speed are improved to a certain extent, which can meet drivers' requirements on driving roads to a large extent and provide a theoretical basis for alleviating traffic congestion.

# LIMITATIONS AND FUTURE WORK

First, the modified model based on the NaSch model proposed in this article lacks validation and calibration under real-time traffic conditions. At present, only the theoretical model has been improved and simulated, and the model needs real-time traffic flow verification and data collection to be more perfect.

Second, the study of road traffic flow is an extremely complex issue. This article only studies and discusses the road model under single-lane conditions. Traffic flow problems such as driving rules under multi-lane changing conditions, drivers' psychological influences, weather conditions, and manual-automatic mixed traffic flow under vehicle interconnection conditions need further research and discussion.

Third, further validation is needed as to whether the model is applicable to both homogeneous and mixed traffic conditions.

## Funding

This research was funded by the research project of the Scientific Research Start of Doctor of Shanxi Agricultural University (Research on Detection Technology of Agricultural Internet of Things Model, 2017YJ30), the Shanxi Province Association for Science and Technology, service economic integration of science and technology demonstration project in 2020 (Research on application status and development trend of agricultural UAV in Shanxi Province), and the Innovation and Entrepreneurship Training Project for College Students in Shanxi Province (Development of equipment for automatic induction fertilization of crops based on soil three-parameter Sensor, 20210160). The funders had no role in study design, data collection and analysis, decision to publish, or preparation of the manuscript.

## Grant Disclosures

The following grant information was disclosed by the authors:
Scientific Research Start of Doctor of Shanxi Agricultural University.
Research on Detection Technology of Agricultural Internet of Things Model: 2017YJ30.
Shanxi Province Association for Science and Technology, service economic integration of science and technology demonstration project in 2020.
Research on application status and development trend of agricultural UAV in Shanxi Province.
Innovation and Entrepreneurship Training Project for College Students in Shanxi Province.
Development of equipment for automatic induction fertilization of crops based on soil three-parameter Sensor: 20210160.

## Competing Interests

The authors declare there are no competing interests.

## Author Contributions

- Xuefeng Deng conceived and designed the experiments, performed the experiments, analyzed the data, prepared figures and/or tables, authored or reviewed drafts of the article, and approved the final draft.
- Yi Shao performed the experiments, analyzed the data, performed the computation work, prepared figures and/or tables, authored or reviewed drafts of the article, and approved the final draft.
- Jiaxin Song performed the computation work, authored or reviewed drafts of the article, and approved the final draft.
- Hui Wu performed the computation work, authored or reviewed drafts of the article, and approved the final draft.

## Data Availability

The code is available in the Supplemental Files.

## Supplemental Information

Supplemental information for this article can be found online at http://dx.doi.org/10.7717/peerj-cs.1102#supplemental-information.

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
