# Peer review of "Traffic flow simulation of modified cellular automata model based on producer-consumer algorithm"

_PeerJ Computer Science, doi:10.7717/peerj-cs.1102_

## Round 0.1 · original submission · Major Revisions

Your manuscript "Traffic flow simulation of modified cellular automata model based on producer-consumer algorithm" has been reviewed. I am happy to inform you that the reviewers have found it attractive. However, they feel that the paper requires major revisions before publication. Authors are suggested to address and clarify some technical issues as recommended by reviewers, mainly to discuss and highlight this manuscript's novelty and main contributions and thoroughly discuss the simulated examples.

Reviewer 1 ·

Basic reporting

1.) The authors should provide a separate section for literature review and not to merge introduction with a review of past literature.

2.) The literature on cellular automata model seems insufficient. I suggest the authors to include more and latest works on a cellular automata model under mixed traffic conditions especially between the lines 61-83. Please refer to the following works:

• Matcha, B. N. et al. (2020) ‘Simulation Strategies for Mixed Traffic Conditions: A Review of Car-Following Models and Simulation Frameworks', Journal of Engineering, 2020.
doi: 10.1155/2020/8231930.

• Tian, J. et al. (2021) ‘Review of the cellular automata models for reproducing synchronized traffic flow’, Transportmetrica A: Transport Science, 17(4), pp. 766–800. doi: 10.1080/23249935.2020.1810820.

• Tong, X. and Feng, Y. (2020) ‘A review of assessment methods for cellular automata models of land-use change and urban growth’, International Journal of Geographical Information Science, 34(5), pp. 866–898. doi: 10.1080/13658816.2019.1684499.

3.) Please summarize the past literature in a tabular format showing the parameters used in the past models, traffic conditions, whether traditional, connected or mixture of both vehicle types, their shortcomings.

4.) I suggest the authors to improve the grammar throughout the paper, especially in the contribution section in the lines 86-97, 246-252, 314-325 where the English seems ambiguous due to which the authors trying to convey are not clearly understood.

5.) Figure 5 is blurry. Please refine the image.

Experimental design

1.) I suggest the authors to tabulate the parameters used in simulation for both traditional NaSch model and Modified model.

2.) Has this modified cellular automata model tested under real-time traffic conditions? Calibrated and validated with real-time traffic flow?? if so please provide the details of location of study, characteristics of traffic flow, data collection, extraction etc. Will this model be suitable for both homogeneous and Mixed traffic conditions??

Validity of the findings

1.) In your contribution section you have stated that "The analysis of the simulation results provides theoretical guidance for enlarging the area of free flow in road traffic, improving the speed of vehicles, and speeding up the dissipation of congestion." Is your model calibrated and tested under real-time traffic flow scenario??

2.) I suggest the authors to provide a separate section for limitations and future work.

Reviewer 2 ·

Basic reporting

Authors propose the producer – consumer approach for road traffic analysis using a cellular automata-based model.
The document is clearly written, unambiguous and easy to read. The bibliography contains relevant and topical titles in the field. The figures are well labelled and described.

Experimental design

The authors investigations are based on a following the leading car model for road traffic. Methods described have sufficient detail and information to replicate, authors also provide the Matlab program for testing.
Compared to traditional conventional cars with human drivers, the car-following behaviors of autonomous vehicles (AVs) and connected autonomous vehicles (CAVs) would be quite different and hence require additional modeling efforts. Some behaviors should be considered like, acceleration distributions, safety gaps, and reaction times of the subjected following-vehicle under numerous traffic conditions and a fuzzy variable to integrate the human behavior.

Validity of the findings

Authors can provide more information (or clarify some aspects) on several topics:
1. How are the cells in the cellular automaton chosen?
2. What traffic rules are considered?
3. How do these interact with the producer & consumer model?
The aspects should be improved in future.

Additional comments

Although your results are compelling, the data analysis could be improved by integrating traffic rules.

---

## Round 0.2 · Minor Revisions

Your manuscript "Traffic flow simulation of modified cellular automata model based on producer-consumer algorithm" has been reviewed. However, a few typos need to be fixed before accepting it for publication:

1) Some of the references are not well formatted:
2) Some titles are written with capital letters.
3) Some titles do not have all the information, such as volume, pages, and year of publication.

Reviewer 1 ·

Basic reporting

No Comment

Experimental design

No Comment

Validity of the findings

No Comment

Additional comments

The authors have modified the paper as per the comments raised.

Reviewer 2 ·

Basic reporting

The authors, in the revised document, provide enough background information and show how their study fit into the problem of using cellular automata for road traffic modelling. The used references are pertinent and used correctly.
The findings are pertinent to the hypothesis of the study.

Experimental design

The solution based on producer-consumer algorithm is pertinent and significant and is presented clear and in detail. Authors took seriously the observations made in the previous step and add more comments to clearly present their work.

Validity of the findings

The information based on which the conclusions are made, are accessible and are reliable. The Matlab code additionally provided can be used to reproduce the results.

Additional comments

Some of the references are not well formatted:
a. Some titles are written with capital letters.
b. Some titles do not have all the information, such as volume, pages, year of publication.

---

## Round 0.3 · accepted · Accept

Your manuscript entitled "raffic flow simulation of modified cellular automata model based on producer-consumer algorithm " has been accepted for publication.

Reviewer 1 ·

Basic reporting

The authors have addressed all my comments.

Experimental design

The authors have addressed all my comments.

Validity of the findings

The authors have addressed all my comments.